# A *Sinorhizobium meliloti* and *Agrobacterium tumefaciens* ExoR ortholog is not crucial for *Brucella abortus* virulence

**Amanda Castillo-Zeledón**[1], **Nazareth Ruiz-Villalobos**[1], **Pamela Altamirano-Silva**[2], **Carlos Chacón-Díaz**[2], **Elías Barquero-Calvo**[1], **Esteban Chaves-Olarte**[2], **Caterina Guzmán-Verri**[1]*

**1** Programa de Investigación en Enfermedades Tropicales, Escuela de Medicina Veterinaria, Universidad Nacional de Costa Rica, Heredia, Costa Rica, **2** Centro de Investigación en Enfermedades Tropicales, Universidad de Costa Rica, San José, Costa Rica

* catguz@una.cr

## Abstract

*Brucella* is a facultative extracellular-intracellular pathogen that belongs to the *Alphaproteobacteria* class. Precise sensing of environmental changes and a proper response mediated by a gene expression regulatory network are essential for this pathogen to survive. The plant-related *Alphaproteobacteria Sinorhizobium meliloti* and *Agrobacterium tumefaciens* also alternate from a free to a host-associated life, where a regulatory invasion switch is needed for this transition. This switch is composed of a two-component regulatory system (TCS) and a global inhibitor, ExoR. In *B. abortus*, the BvrR/BvrS TCS is essential for intracellular survival. However, the presence of a TCS inhibitor, such as ExoR, in *Brucella* is still unknown. In this work, we identified a genomic sequence similar to *S. meliloti exoR* in the *B. abortus* 2308W genome, constructed an *exoR* mutant strain, and performed its characterization through *ex vivo* and *in vivo* assays. Our findings indicate that ExoR is related to the BvrR phosphorylation state, and is related to the expression of known BvrR/BrvS gene targets, such as *virB8*, *vjbR*, and *omp25* when grown in rich medium or starving conditions. Despite this, the *exoR* mutant strain showed no significant differences as compared to the wild-type strain, related to resistance to polymyxin B or human non-immune serum, intracellular replication, or infectivity in a mice model. ExoR in *B. abortus* is related to BvrR/BvrS as observed in other Rhizobiales; however, its function seems different from that observed for its orthologs described in *A. tumefaciens* and *S. meliloti*.

## Introduction

Members of the *Brucella* genus are worldwide distributed zoonotic pathogens that belong to the *Alphaproteobacteria* class. Infection in humans is associated with diverse and unspecific symptoms, including fever, lassitude, general malaise, weight loss, headache, low back pain, and arthralgias [1]. *Brucella abortus* infects bovines, where it preferentially replicates in reproductive organs, causing abortion, infertility, decreased milk production, reproductive failure,

**Data Availability Statement:** All relevant data are within the manuscript and its Supporting information files.

**Funding:** This work was supported by Fondos del Sistema FEES/CONARE [02-2020, 0652-19 to C. G-V], Fondos FIDA, Universidad Nacional [SIA 0047-17 to C. G-V], Espacio Universitario de Estudios Avanzados, UCREA [B8762] from the Presidency of University of Costa Rica, and the Vice Presidency for Research, University of Costa Rica [C0456 to E.C-O]. The funders had no role in study design, data collection, and analysis, decision to publish, or preparation of the manuscript.

**Competing interests:** The authors have declared that no competing interests exist.

and epididymitis [2, 3]. As a result, countries with the disease go through economic losses in the animal industry and public health [4].

*B. abortus* is a facultative extracellular-intracellular pathogen [4, 5], hence the transition from lifestyles requires a fine-tuned transcriptional regulatory network coordinating a bacterial response to environmental signal cues [6]. Lifestyles transitions have been linked with an invasion switch pathway for *Sinorhizobium meliloti*, a plant symbiont, and *Agrobacterium tumefaciens*, a plant pathogen [7–9]. *S. meliloti* invasion switch pathway is known as RSI (Exo**R**, Exo**S**, and Chv**I**). It involves a two-component system (TCS), ExoS/ChvI, and a periplasmic protein, ExoR. ExoS is a membrane-bound sensory protein with histidine kinase activity, and ChvI is its cognate response regulator [10, 11]. According to the canonical TCS OmpR model [12], it is inferred that ExoS is autophosphorylated upon sensing an external stimulus. Subsequently, the phosphate group is transferred to ChvI, increasing its affinity for regulatory DNA sequences, hereafter influencing the transcription of genes involved in responding to the original external stimulus. ExoS/ChvI activation is required to transition from a flagellum producing free-living cell to a succinoglycan producing host-invading cell. Constitutive *exoS* mutants are non-motile, do not harbor flagella, overproduce succinoglycan, and form loose biofilms [10, 13, 14]. The third member, ExoR, is a periplasm protein that contains Sel1-like repeats that mediate protein-protein associations. It is known to interact with ExoS in the periplasm, preventing ExoS/ChvI activity. Inhibition of ExoS/ChvI activity induces free-living forms with flagella and no succinoglycan production. These free-living forms cannot induce root nodulation and plant cell colonization [15–17]. ExoR undergoes proteolysis by unknown mechanisms and relieves the TCS suppression [18].

*A. tumefaciens* invasion switch pathway is termed RGI (Exo**R**, Chv**G**, Chv**I**) [9, 19]. In these bacteria, the plant cell invasion mechanism requires attachment to the host cell and biofilm formation. Without ExoR or its proteolysis induced at low pH, *A. tumefaciens* exhibit a non-motile, hypermucus phenotype and disabled surface adherence. Consequently, plant colonization fails [9, 20]. More than 400 differentially expressed genes have been found in mutants lacking *exoR* and for this reason, it is cataloged as a global regulator [20].

Phylogenetic distribution analysis of the invasion switch pathway proteins in 57 *Alphaproteobacteria* genomes found putative orthologs, including one in *B. abortus* [7]. The TCS BvrR/BvrS in *B. abortus* was first discovered in 1998 and is phylogenetically related to ExoS/ChvI and ChvG/ChvI [21]. BvrS, the sensory protein, does not sense a unique signal but rather an environmental cue constituted of different factors, including low pH and nutrient-limiting conditions, probably similar to those found during *B. abortus* trafficking through autophagosome-like compartments [22–24]. Its activation promotes the signal transference to BvrR, the regulatory protein [21]. The *bvrR* or *bvrS* mutants have increased susceptibility to polymyxin B and non-immune serum and are attenuated in mouse and cell models [25]. This phenotype was attributed to changes in cell envelope homeostasis and regulation of carbon and nitrogen metabolism [26].

Transcriptomic and proteomic analyses of the *B. abortus* null *bvrR* and *bvrS* mutants have shown more than 100 differentially transcribed genes and expressed proteins as compared to the wild-type [27, 28]. Direct binding of BvrR to regulatory regions of *virB* shows that BvrR/BvrS influences *virB* transcription, encoding a type IV secretion system, crucial for bacterial intracellular survival [22, 29]. For other proteins, as the outer membrane protein (OMP) Omp25 (also known as Omp3a) and the transcriptional regulator VjbR, evidence indicates that BvrR/BvrS positively affects their transcription since Omp25 and VjbR are not expressed when BvrR or BvrS are absent [29, 30]. Additionally, a direct interaction between BvrR and the upstream region of VjbR was evidenced [22, 29]. VjbR belongs to the LuxR regulator family of

transcriptional activators related to quorum sensing [31–33]. Furthermore, direct regulation of VjbR over the *virB* promoter has been demonstrated [22, 29].

To date, ExoR has not been described for *B. abortus* or any other *Brucella* spp. Due to the homology between ExoS and BvrS, it is hypothesized that the *Brucella* ExoR ortholog could be functionally related to BvrR/BvrS [28]. To elucidate this hypothesis, an *exoR* mutant derived from the reference strain *B. abortus* 2308 Wisconsin was constructed and studied [34]. The results show that ExoR affects VirB, Omp25, and VjbR expression dynamics and BvrR phosphorylation, but its absence does not affect *B. abortus* virulence *ex vivo* or *in vivo*.

## Results

### *B. abortus* 2308W genome encodes an *exoR* ortholog

Using Basic Local Alignment Search Tool (BLASTp) and the ExoR protein sequence from *S. meliloti* (GenBank Accession WP_003534542.1) [35, 36] a similar sequence (51% identity), possibly encoding a 267 amino acid residues long protein was found in the *B. abortus* 2308W genome (GenBank Accession ERS568782) [34]. This sequence was found under locus tag BAW1_0856, between open reading frames annotated as a predicted protein and an exodeoxyribonuclease III. Based on this information, we constructed a *B. abortus* 2308W mutant with an in-frame deletion of 171 internal amino acid residues. Deletion of the 515 bp was confirmed by PCR, Southern blot assays [37], and Sanger sequencing, as described in the Methods section and S1 Fig. No significant differences in growth kinetics in TSB (Fig 1A) and sugar assimilation, according to API 50 CH assay, were detected between the mutant and the wild-type strain.

**The *exoR* mutation does not affect *B. abortus* resistance to polymyxin B and non-immune human serum.** The *B. abortus* LPS smooth phenotype was not altered by the *exoR* mutation, as judged by the acriflavine agglutination test. Exposure to non-immune human serum and increasing polymyxin B concentrations at pH 7 and pH 6 did not reveal differences between the wild-type and the *exoR* mutant strain (Fig 1B, 1C and 1D). As expected, the *bvrR* and *bvrS* mutants, included as controls, were sensitive to polymyxin B and non-immune human serum (Fig 1C and 1D). Both strains showed significant differences when compared to the wild-type and the *exoR* mutant.

**ExoR impacts the expression dynamics of proteins related to BvrR/BvrS in nutrient rich conditions.** We assessed the expression of BvrR and BvrS by western blot of cell lysates obtained at different moments of growth in TSB. We observed similar BvrR and BvrS expression patterns compared to the wild-type strain in the mutant strain (Fig 2). The BvrR phosphorylation state was analyzed by Phos-tag SDS-PAGE and western blot [22, 38]. As expected, two bands were recognized, corresponding to BvrR-P (upper band) and non-phosphorylated BvrR (Fig 2). BvrR-P remained constant in the wild-type strain over time, ranging from 11 to 20% of total BvrR, until 28 hours of growth where there was no phosphorylation signal. In the *exoR* mutant, BvrR-P constantly decreased, and its signal disappeared more radidly, going from 15 to 6%, and being undetectable at 24 hours.

We also evaluated three proteins whose expression is positively regulated by BvrR/BvrS: VirB8, VjbR, and Omp25 (Fig 3). VirB8 and VjbR reached a maximum expression around late-log and mid-log in the wild-type strain, respectively, and then decreased until reaching the stationary phase, as previously reported [29]. In the *exoR* mutant, VirB8 and VjbR expression showed a similar trend; nevertheless, the expression declined before it occurred in the wild-type, around mid-log. Omp25 showed slight differences in expression according to the wild-type strain time of growth as reported [39]. In the *exoR* mutant, expression levels gradually decreased from the early log until the stationary phase of growth.

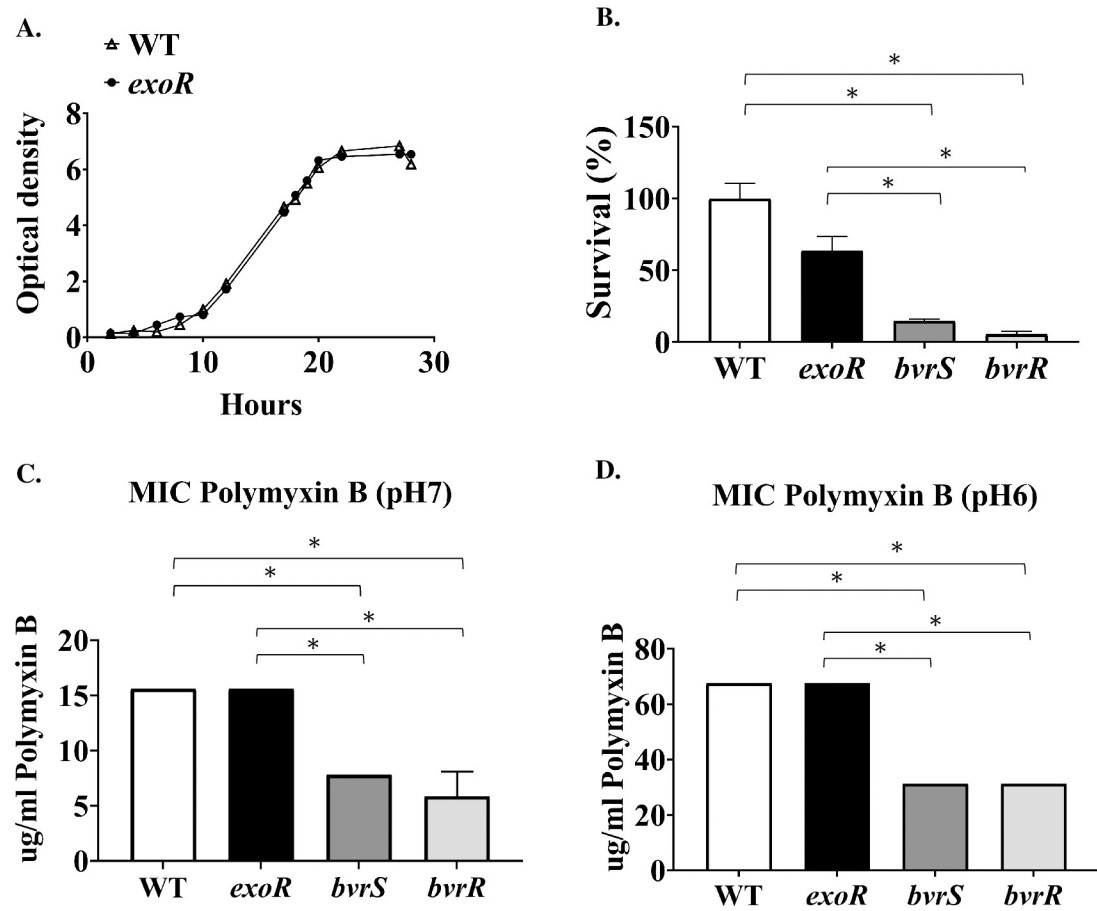

**Fig 1. The *exoR* mutation does not alter bacterial growth, resistance to non-immune human serum or polymyxin B treatment.** (A) *B. abortus* 2308W (WT) and the *exoR* mutant (*exoR*) were grown in TSB at 37˚C, and the absorbance was measured at 420 nm at the indicated times. (B) The indicated strains *B. abortus* 2308W (WT), *exoR*, *bvrS*, and *bvrR* mutants, were exposed to non-immune human serum at 37˚C for 90 min. These experiments are representative of at least three performed. (C) and (D) The indicated strains, *B. abortus* 2308W (WT), *exoR*, *bvrS*, and *bvrR* mutants, were exposed to increasing concentrations of polymyxin B for 48 hours at 37˚C, at pH 7 and pH 6, respectively. The minimum inhibitory concentrations shown are representative of at least three experiments performed. The *bvrS* and *bvrR* mutant strains were used as susceptibility controls. *, $P < 0,05$ (Anova and Kruskal-Wallis test for multiple comparisons).

**Low nutrients and pH do not significantly alter *B. abortus* exoR phenotype as compared to nutrient rich conditions.** BvrR phosphorylation increases when exponential growing bacteria are exposed to minimal medium and pH 5.0 [22]. These conditions mimic the intracellular environment, where BvrR/BvrS is critical for survival [23, 24, 40]. A possible role for ExoR related to the activation of BvrR/BvrS was assessed using similar assays. Hence, the polymyxin B and non-immune human serum assays were carried out using bacteria after 4-hours incubation in minimal medium and pH 5.0 (Fig 4A and 4B). No significant difference was found between the wild-type and the mutant strain. As expected, the *bvrR* mutant, included as a control, was susceptible to polymyxin B and non-immune human serum. This strain showed significant differences when compared to the wild-type strain and the *exoR* mutant.

The expression of VirB8 and VjbR in the wild-type strain under these conditions was similar to the previously reported and related to BvrR transient phosphorylation [29]. The *exoR* mutant showed a similar expression pattern as compared to the wild-type for both proteins

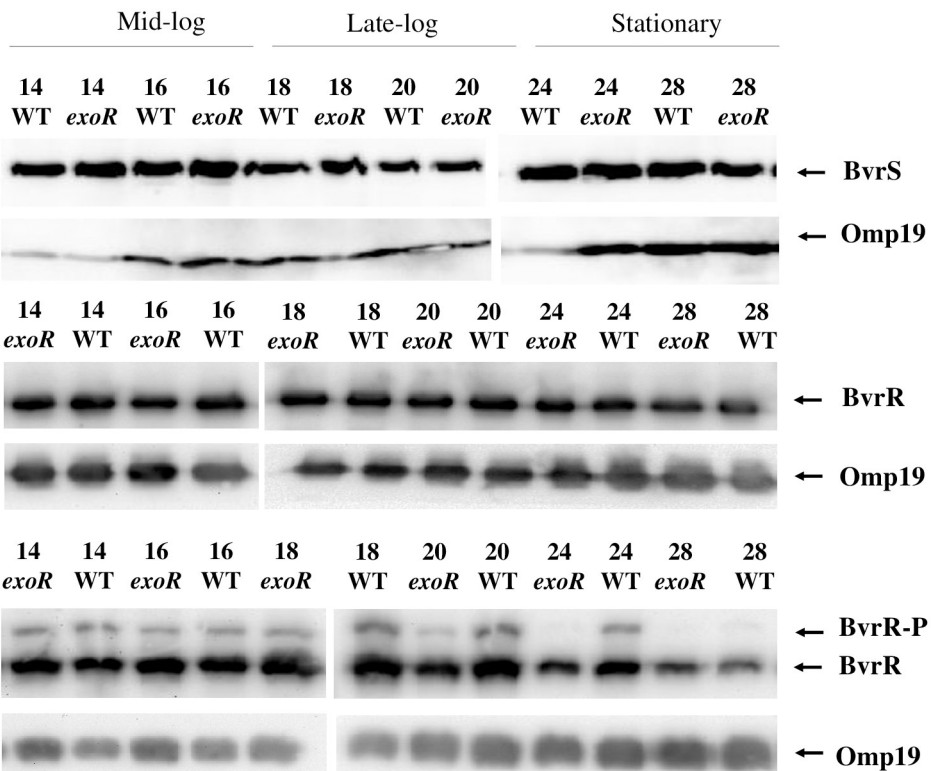

**Fig 2. BvrR/BvrS expression and BvrR phosphorylation dynamics in the *exoR* mutant.** Western blot analysis of BvrS, BvrR, and BvrR-P expression according to growth phase in the *B. abortus exoR* mutant (*exoR*) and *B. abortus* 2308W (WT). Both strains were grown in TSB for 48 hours. Representative hours of different growth phases were analyzed (mid-, late-log, and stationary) indicated on top. For western blot, equal amounts (20 µg) of whole-bacterium lysates were separated by 12,5% SDS—PAGE. For phosphorylation analyses, samples were separated by 10% SDS-PAGE containing Phos-tag. PVDF membranes were incubated with anti-BvrS, anti-BvrR, and and reprobed with anti-omp19 antibodies indicated on the right side. After incubation, the immune complexes were detected by chemiluminescence reaction. Omp19 was used as a loading control. The blots for BvrS, BvrR and Phos-tag are from independent gels.

during the time points tested, however, their expression seems lower, particularly for VirB8, except for the 6h time point (Fig 4C).

**The *exoR* mutant successfully infects mice and cell cultures.** The ability to escape the host immune response and to survive intracellularly is critical for *B. abortus* pathogenicity [41, 42]. Cellular and murine models were used to evaluate the *exoR* mutant strain infective abilities. Fig 5A and 5B show that the *exoR* mutant infected HeLa and Raw macrophage cells similar to the wild-type strain. Fig 5C shows that the dynamics of infection of the wild-type strain in mice were consistent with previous results [26, 43]. We found no significant differences between the wild-type and the *exoR* mutant, indicating that ExoR is not required for *B. abortus* infection in mice or cell culture.

## Discussion

BvrR/BvrS homologs in *S. meliloti* and *A. tumefaciens* are required to associate with plant cells [14, 36]. Similarly, *B. abortus* needs BvrR/BvrS to infect the host animal cell successfully [44]. However, the RSI invasion switch functioning in both plant cell microbes, where ExoR is a protagonist, seems dissimilar in *B. abortus*.

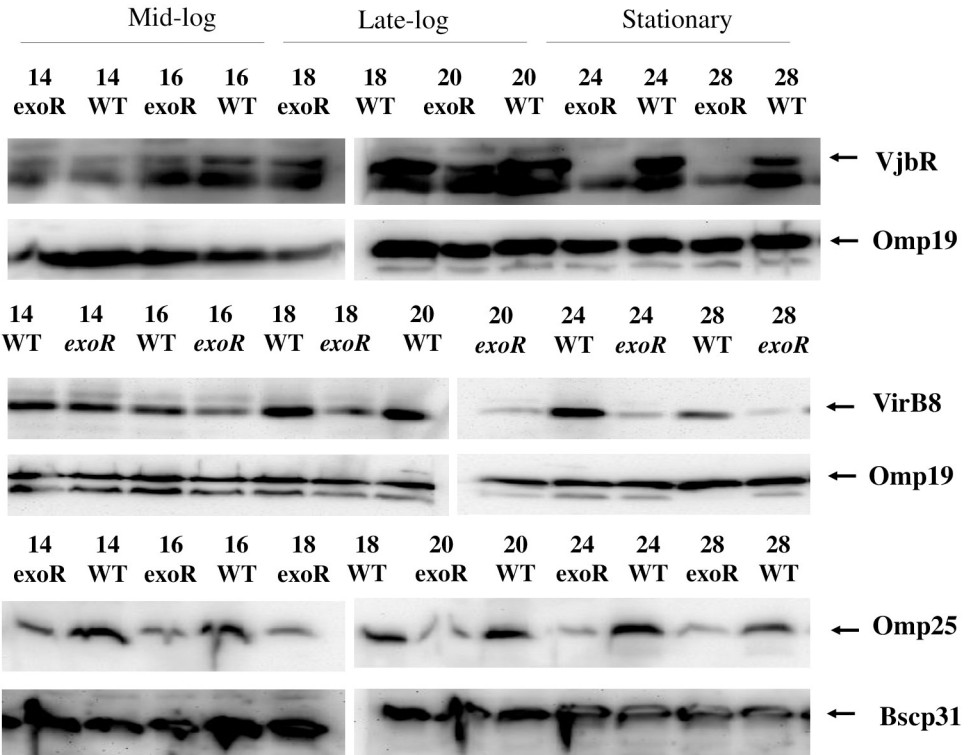

**Fig 3. VjbR, VirB8, and Omp25 expression dynamics are different in the *exoR* mutant as compared to the wild-type strain.** Western blot analysis of VirB8, VjbR, and Omp25 expression according to growth phase in the *B. abortus exoR* mutant (*exoR*) and *B. abortus* 2308W (WT). Both strains were grown in TSB for 48 hours. Representative hours of different growth phases were analyzed (mid-, late-log, and stationary) indicated on top. For western blot, equal amounts (20 μg) of whole-bacterium lysates were separated by 12,5% SDS—PAGE. PVDF membranes were incubated with anti-VirB8, anti-VjbR, anti-Omp25, and then reprobed with anti-Omp19 or anti-Bscp31 antibodies, indicated on the right side. After incubation, the immune complexes were detected by chemiluminescence reaction. Omp19 and Bscp31 were used as loading control. The blots for VirB8, VjbR and Omp25 are from independent gels.

In the RSI invasion switch, ExoR levels are critical for host interaction and subsequent invasion in *S. meliloti. exoR95*:: Tn*5* loss-of-function mutant overproduces succinoglycan and has a dramatic decrease of flagellar gene expression and other genes related to motility and chemotaxis [18, 36]. A similar effect was observed in an *A. tumefaciens exoR* null mutant [9]. In both cases, host interactions are compromised. According to our mice model result, this does not seem to be the case for *B. abortus*.

However, ExoR seems to be related to BvrR/BvrS. Proteins whose gene expression is under the control of BvrR/BvrS [29, 45] showed a different expression pattern in the *exoR* mutant as compared to the wild-type strain. The transcriptional regulator VjbR, the T4SS VirB component VirB8 (essential for intracellular survival) [23, 29, 46, 47] and Omp25 (with a potential role in virulence) [48–50], were expressed in the Δ*exoR* mutant at lower levels, particularly starting at late log growth phase, as compared to the wild strain. Nevertheless, the *exoR* mutant was able to infect and survive in cell and mice models. These results were unexpected considering the high degree of identity between the BvrR, BvrS, and ExoR homologs. If the *B. abortus* ExoR homolog is functioning as its counterparts, i.e., as a sensor protein inhibitor, the TCS's upregulated genes would be overexpressed. For example, in *A. tumefaciens*, ChvG/ChvI upregulates the expression of an Omp25 orthologue, known as AopB [20, 38]. This protein

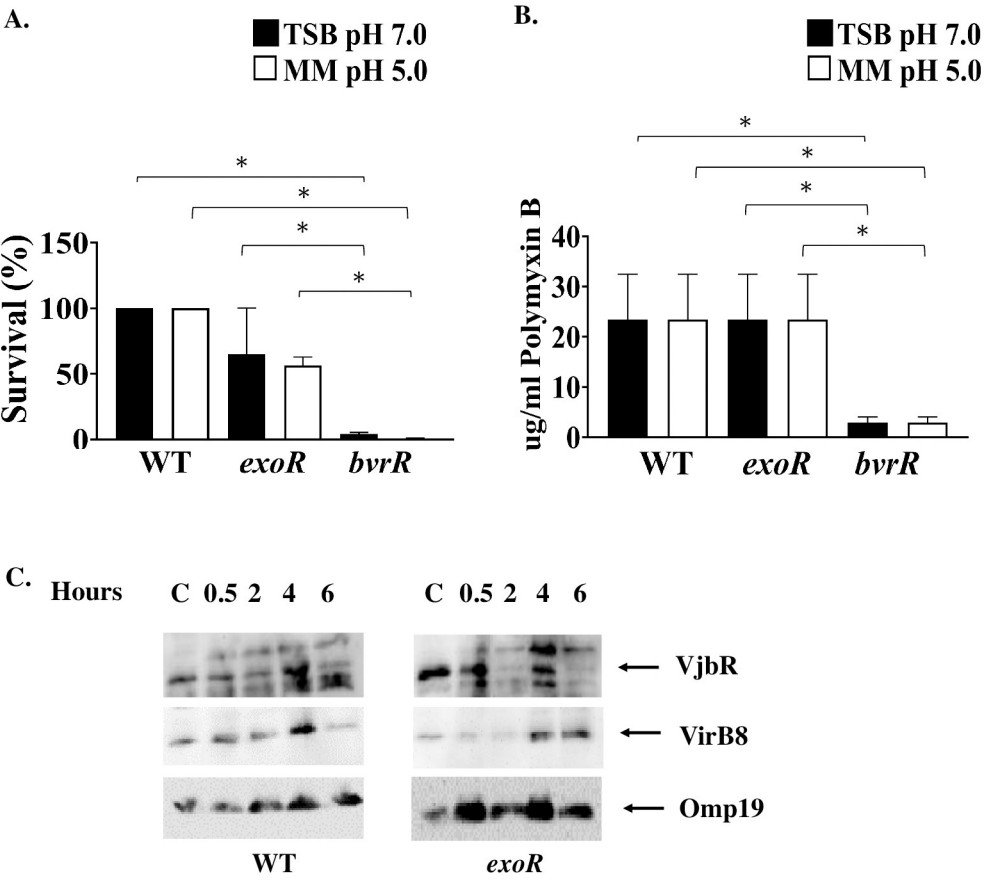

**Fig 4.** ***In vitro* conditions mimicking the intracellular environment do not affect *B. abortus exoR* resistance to non-immune human serum or polymyxin B treatment and induces expression of VjbR and VirB.** (A) *B. abortus* wild-type 2308W, *B. abortus exoR*, and *B. abortus bvrR* (negative control) mutant strains were grown to exponential phase and incubated in minimal medium (MM) at pH 5.0 or rich medium (TSB) at pH 7.0 for 4 hours. After incubation, bacteria were exposed to non-immune human serum at 37˚C for 45 min or (B) exposed to increasing concentrations of polymyxin B for 48 hours at 37˚C at pH 7. The Anova and Kruskal-Wallis test for multiple comparisons was used. *, $P < 0,05$. (C) The wild-type and *exoR* mutant were grown as indicated in (A) for 6 hours, and 1 mL culture of each strain was resuspended in minimal medium, pH 5 for 0.5, 2, 4, and 6 hours. After incubation, lysates were prepared, separated by 10% SDS-PAGE and assessed using anti-VjbR and anti-VirB8 antibodies by western blotting. Omp19 detection was used as a loading control. For comparison, a bacterial sample from each strain, with no exposure to minimal medium, pH 5 was included and is labeled as "C".

influences the stability, permeability, and topology of the membrane, as does Omp25 and other OMPs in *B. abortus* [51, 52]. Since ExoR represses ChvG by direct interaction, inhibiting the signal transferred through the TCS, *aopB* was highly upregulated in the *A. tumefaciens exoR* null mutant [20]. The same has been observed for succinoglycan genes that are upregulated by ExoS/ChvI from *S. meliloti* [13]. Surprisingly, in *B. abortus exoR* mutant Omp25 was down expressed. Moreover, BvrR and BvrS did not show differences in their expression compared to the wild-type. This suggests that their expression is independent of ExoR, but possible posttranslational modifications (including phosphorylation and dephosphorylation) might not, since the BvrR phosphorylation dynamics was different as compared to the wild-type strain. The *exoR* mutant showed a gradual fade of the BvrR-P signal during bacterial growth, which occurs faster than in the wild-type strain. There is no direct evidence linking the phosphorylation of the response regulator and ExoR in other bacterial RSI switches; but, in *A.*

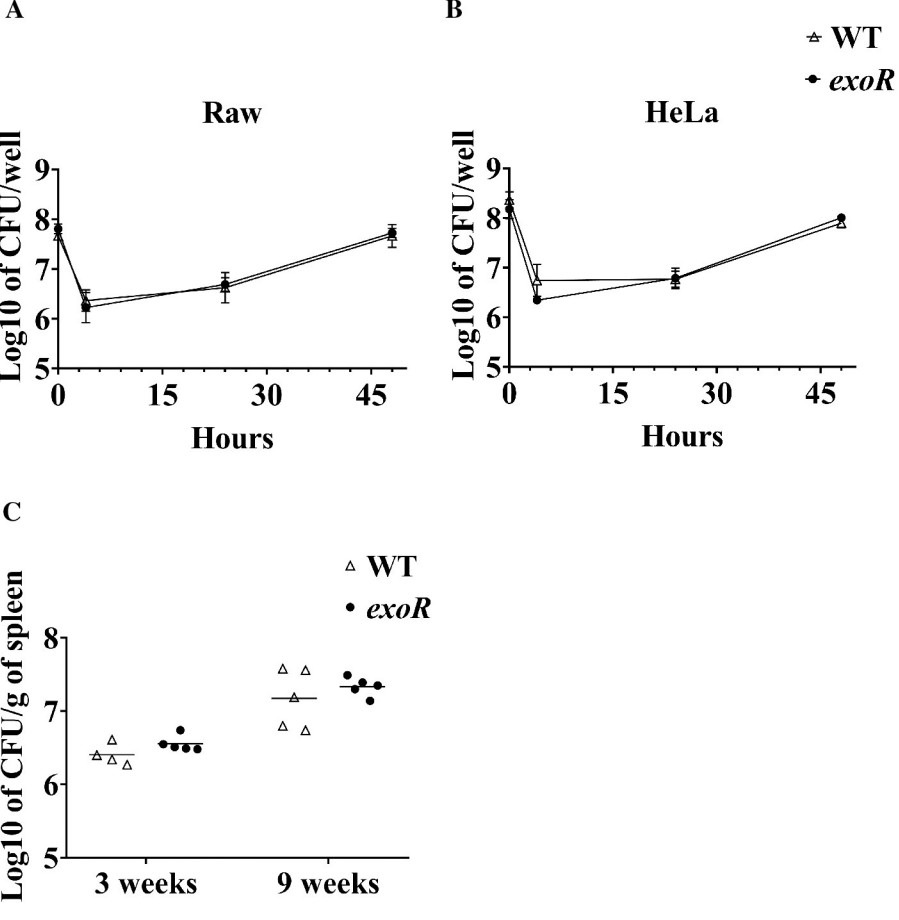

**Fig 5. The *exoR* mutation in *B. abortus* 2308W does not affect its infection capacity.** (A, B) Murine RAW 264.7 macrophages or HeLa epithelial cells were cultivated and infected with *B. abortus* strains (2308W and *exoR* mutant) in the exponential phase. The number of intracellular viable *B. abortus* CFU was determined at different hours post-infection. Each point is the average of three independent trials. (C) Virulence assays of *B. abortus* 2308W and *exoR* mutant in mice. Mice were intraperitoneally inoculated with $10^6$ CFU of either *B. abortus* 2308W or *B. abortus* 2308W *exoR* mutant. CFU per gram of spleen was determined at 3- and 9-weeks post-infection. For these experiments, no significant difference was shown between the wild-type and the mutant (Mann-Whitney test).

*tumefaciens* and *S. meliloti* ExoR absence induces a phenotype similar to that of bacteria with a constitutive TCS. This does not seem to be the case in *B. abortus* 2308W.

In *A. tumefaciens* ChvG/ChvI is activated under low-pH conditions and ExoR represses it under neutral conditions. A null *exoR* mutant shows a phenotype at neutral pH similar to that of acid-exposed wild-type bacteria, demonstrating that ExoR is required to inactivate the system [20]. In *S. meliloti*, ExoR inhibits flagella production and stimulates succinoglycan production. Mutants with defective ExoR mimic non-motile succinoglycan-producing bacteria, demonstrating the need for ExoR to inhibit ExoS [18]. This inhibitory ExoR function on BvrR/BvrS was not clearly evident in our mutant under neutral pH conditions. Therefore, we also evaluated the possibility that the *exoR* mutation has an evident effect under starving and acid-exposed conditions, which are also relevant for BvrR phosphorylation, and induction of VjbR and VirB expression in the wild-type strain [22]. Our results show that, as in rich medium, pH 7, the *exoR* mutation has no effect after exposure to minimal medium and pH 5, on the *B.*

*abortus* ability to resist polymyxin B or non-immune human serum treatment, and that there are subtle differences in expression of VjbR and VirB8 as compared to the wild-type strain.

Nevertheless, there seems to be a partial functional similarity between ExoR from *B. abortus* and its orthologs. ExoR is not required for *B. abortus* virulence in the analyzed models. However, there is a functional link with BvR/BvrS, clearly shown by altering BvrR phosphorylation patterns and expression levels of some of the downstream BvrR/BvrS targets. One possible interpretation is that ExoR is inhibiting, directly or indirectly, the dephosphorylation of BvrR and consequently, the expression of the analyzed genes. Unlike the studies published for *S. meliloti* and *A. tumefaciens*, we conclude *B. abortus exoR* mutant maintains the ability to invade and replicate inside cells and survive inside the host. Since the introduced deletion in *exoR* did not remove the gene entirely, we cannot exclude that the remaining protein segment kept some of the functions of the native ExoR. More studies are needed to understand the role of the conserved ExoR, not only in *B. abortus* but also in other members of the genus, as well as other members of the *Alphaproteobacteria*.

## Methods

### Bacterial culture

All procedures involving live *B. abortus* were carried out according to the "Reglamento de Bioseguridad de la CCSS 39975–0", 2012, after the "Decreto Ejecutivo #30965-S", 2002 and research protocol SIA 0652–19 approved by the National University, Costa Rica. *B. abortus* 2308W (WT *Brucella* strain, virulent, smooth LPS, NaI$^r$) [53] or *B. abortus* 2308W *exoR* mutant strain (this study) were maintained and grown at 37˚C with agitation (200 rpm) in standard Tryptic Soy Broth (TSB) at pH 7.0 or minimal medium (33 mM KH$_2$PO$_4$, 60.3 mM K$_2$HPO$_4$, and 0.1% yeast extract) at pH 5.0 (adjusted with citric acid). *Escherichia coli* S17-1 λpir strain (*sup E44*, *recA1*, *endA1*, *hsdR17*, *thi-1*, *gyrA96*, *relA1*, lysogenic phage λpir) and *E. coli* Top10 (F- *mcrA* Δ (*mrr-hsd*RMS-*mcr*BC) Φ80*lac*ZΔM15 Δ *lac*X74 *rec*A1 *ara*D139 Δ (*ara-leu*)7697 *gal*U *gal*K *rps*L (StrR) *end*A1 *nup*G) (Invitrogen™) used for cloning were grown in LB (37˚C) (Simon et al., 1983). When needed, *B. abortus* and *E. coli* strains were supplemented with antibiotics to maintain plasmid selection.

### Strain construction

*B. abortus exoR* mutant was constructed as reported elsewhere [54, 55]. Briefly, in-frame deletion was generated by PCR overlap using genomic DNA of *B. abortus* as a template. Primers were designed using the available genome sequence ERS568782, corresponding to reference strain *B. abortus* 2308W to perform a 515 bp internal deletion (position +172 to +687 from the first codon ATG), leaving only 35% of the coding sequence (S1 Fig). Final validation of successful deletion was done using Sanger DNA sequencing of selected fragments and Southern blot (see below and S1 Fig).

Plasmid and chromosomal DNA were extracted with QIAprep Spin Miniprep and DNeasy Blood and Tissue Kit (Qiagen).

### Carbohydrate assimilation pattern

The carbohydrate assimilation pattern was performed using the API 50 CH kit (Biomerieux) according to the manufacturer's instructions. Briefly, a bacterial suspension of both strains, mutant and wild-type, grown in TSB at pH 7, with agitation of 200 rpm to exponential phase, were used to rehydrate each of the wells. The strips were incubated at 37˚C for 18 hours. During incubation, metabolism produces color changes. Positive and negative test results were

annotated to obtain a profile. The profile from three different assays was compared to the wild-type strain profile.

## Southern blot

Whole genomic DNA (0.5 μg) from *Brucella* strains was digested with 5U of the restriction endonucleases BamHI and PstI (Fermentas) in a reaction volume of 20 μl and according to the manufacturer's instructions. The resulting fragments were separated in a 0.7% agarose gel. After electrophoresis, hybridization was carried out with minor variations as described previously [37]. Primers bruabI0884.3 and bruabI0884.5 were used to generate a 295 bp amplicon from *B. abortus* 2308W DNA (S1 Table). These primers amplify a region from position 68 bp to -363 from *exoR* first codon (S1 Fig), used as a probe after digoxigenin labeling (DIG DNA Labeling and detection kit, Roche). The labeled probe was denatured and added to the hybridization buffer. Hybridization was performed overnight. After exposure to ultraviolet light, the membrane was then washed, blocked, incubated with anti-DIG alkaline phosphatase-conjugated FAB-antibody, and washed according to the manufacturer's instructions. Luminescence was recorded on X-Ray films (Kodak) for 4 hours.

## Polymyxin B sensitivity assay

Polymyxin B (Sigma, USA) sensitivity assays were performed in triplicate as described with some modifications [56]. Both strains, mutant, and wild-type, were grown until exponential phase in TSB at pH 7, with the agitation of 200 rpm. Bacteria of each strain were adjusted to $5x10^6$ CFU/mL in TSB. 100 μL sample of each strain was mixed with 100 μL of different polymyxin B concentrations in a microplate (the final concentration of the first well was 250 ug/mL). After 48 h of incubation at 37˚C in two different pH (6 and 7), the minimum inhibitory concentration for polymyxin was calculated.

## Sensitivity to the bactericidal action of non-immune serum

Exponentially growing bacteria grown in TSB at pH 7, with agitation of 200 rpm, were adjusted to $10^4$ CFU/mL in PBS and dispensed in duplicate in microtiter plates (200 μl per well) containing fresh human serum (400 μl/well). After 90 min of incubation at 37˚C, 100 μl was plated on tryptic soy agar. Results were expressed as the percentage CFU as compared to controls performed with decomplemented serum at room temperature for 30 min. To assess complement consumption, serum was homogenized with dehydrated yeast *Saccharomyces cerevisiae* for 1 h at 37˚C. After incubation, the serum was centrifuged at 14000 rpm for 5 min [25].

## Western blot and BvrR phosphorylation

Detection of BvrS, BvrR, VirB8, VjbR, Omp25, and determination of BvrR phosphorylation was performed as previously described [22]. Bacteria grown to exponential phase in TSB at pH 7, with the agitation of 200 rpm, were concentrated by centrifugation, resuspended in Laemmli sample buffer, and heated at 100˚C for 20 min. The protein concentration was determined by the Bio-Rad DC method according to the manufacturer's instructions. Equal amounts of protein (20 μg) were loaded onto a 10% gel for SDS-PAGE. Separated proteins were transferred to a polyvinylidene difluoride (PVDF) membrane and probed with the indicated antibodies. To analyze the phosphorylated status of BvrR, samples were solubilized in Laemmli sample buffer without heating, and equal amounts of protein (20 μg) were loaded onto a 10% gel for SDS-PAGE containing Phos-tag (100 mM) and MnCl2 (0.2 mM). Recombinant BvrR

phosphorylated with the phosphate universal donor carbamoyl phosphate was used as a positive control. The percentage of BvrR-P from total BvrR was calculated for each indicated condition by densitometry from at least three independent experiments.

For loading control each membrane was reprobed. First, the membranes were washed with PBS Tween 0,1% for 30 minutes and incubated in glycine 0.1M pH 2.5 for 1 hour. After incubation, membranes were probed with Omp19 or Bscp31 antibody.

## Exposure of *Brucella* strains to minimal and low pH medium

Bacterial strains were grown *in vitro* in TSB to exponential growth phase [22]. A volume of each culture corresponding to $10^4$ CFU/mL or $5x10^6$ CFU/mL was calculated for the sensitivity to the bactericidal action of non-immune serum or the polymyxin sensitivity assays, respectively. Bacteria were centrifuged at 10 000 x *g* for 3 min and resuspended in minimal medium at pH 5.0 or in a nutrient-rich medium (TSB) at pH 7.0 for 4 h at 37˚C at 200 rpm. After incubation, bacteria were concentrated by centrifugation at 10,000 x *g* for 3 min and tested as described above. For the VirB8 and VjbR protein expression assays, aliquots of 1mL of bacteria grown in TSB to exponential growth phase were centrifuged, resuspended in minimal medium, pH 5 for 0.5, 2, 4, and 6 hours. After incubation, each sample was centrifuged and tested as described above. For comparison, a 1mL aliquot with no exposure to minimal medium, pH 5 was included.

## Gentamicin-protection assay and intracellular replication quantification

Murine RAW 264.7 macrophages (ATCC TIB-71) or HeLa epithelial cells (ATCC clone CCl-2) were cultivated and infected with *B. abortus* strains at exponential and stationary growth phase as previously described with some modifications [22]. Cells were seeded in 24-well tissue culture plates and multiplicities of infection (MOI) of 100 for macrophages and 500 for HeLa epithelial cells. The number of intracellular viable *B. abortus* CFU was determined at different hours post-infection. Cells were washed twice with phosphate-buffered saline (PBS) and treated with Triton X-100 (0.01%). Lysates were serially diluted and plated on tryptic soy agar dishes for the quantification of CFU.

## Virulence assays in mice

BALB/c female (18–24 grams) mice were intraperitoneally (i.p.) inoculated with $10^6$ UFC of either *B. abortus* 2308W or *B. abortus* 2308W *exoR* mutant at exponential phase grown in TSB at pH 7, with agitation of 200 rpm. Mice were sacrificed at 3- and 9-weeks post-infection. Spleen counts were determined as described elsewhere [57]. Protocols of experimentation were revised and approved by the Welfare Commission of the Veterinary School at Universidad Nacional, Costa Rica, under protocol number FCSA-EMV_CBAB-009-2015, and agreed with the corresponding law, Ley de Bienestar de los Animales, of Costa Rica (law 7451 on animal welfare). Only certified veterinarians carried out these experiments and all efforts were made to minimize suffering.

## Statistical analysis

Statistical analyses were performed using GraphPad Prism software. All results are presented as means ± SD from at least three independent experiments unless otherwise stated. Non-parametric statistics were used. Anova and Kruskal-Wallis test were used for multiple comparisons and Mann-Whitney test for the comparison of two strains.

## Supporting information

**S1 Fig. Mutant construction and Southern blot.** (A) Schematic representation of the in-frame deletion strategy used to construct the *exoR* mutant. Primers exoR-F1 and exoR-R2 were used to generate fragment 1, and fragment 2 was generated using exoR-F3, and exoR-R4 (S1 Table). Both fragments were ligated by PCR overlapping using nucleotides exoR-F1 and exoR-R4. The resulting deletion allele was cloned in the pCR™ 2.1 vector (Invitrogen™) and subcloned into the BamHI-XBaI site of the suicide plasmid pJQKm. Plasmid pJQKm containing the deleted allele was introduced in *B. abortus* 2308W by conjugation. Colonies corresponding to integrating the suicide vector in the chromosome were selected using Nalidixic acid (NaI, 25 µg/mL) and Kanamycin (Km, 50 µg/mL) resistance. Excision of the suicide plasmid leading to the mutant's construction by allelic exchange was selected by 5% sucrose resistance and Km sensitivity. The resulting colonies were screened using primers exoR-F1 and exoR-R4. Mutant colonies generated a 780 bp fragment and the parental strain a 1200 bp fragment. (B) Southern blot analysis. BamH1 and PstI restriction sites were chosen according to the *B. abortus* 2308W genome sequence, at positions -3009 and +1552, respectively, from *exoR* first codon. The resulting fragments were separated in a 0.7% agarose gel. After electrophoresis, the protocol was carried out with minor variations as described in Methods. Briefly, the gel was rinsed in 0.25 M HCL, denatured (1.5 M NaCl, 0.5 M NaOH), and neutralized twice (1.5 M NaCl, 0.5M Tris HCl pH8). The DNA was then transferred to a nylon membrane (Roche) overnight in 10x SSC pH 7 (0.15 M sodium citrate, 1.5 M NaCl). Primers bruabI0884.3 and bruabI0884.5 (S1 Table) were used to generate a 295 bp amplicon from *B. abortus* 2308W DNA. These primers amplify a region from position 68 bp to -363 from *exoR* first codon. The amplicon was purified from agarose gels using QIAquick Gel Extraction Kit (Qiagen) and labeled with digoxigenin for use as a probe according to the manufacturer's instructions (DIG DNA Labelling and detection kit, Roche). The labeled probe was denatured and added to the hybridization buffer. Hybridization was performed overnight. After exposure to ultraviolet light, the membrane was then washed, blocked, incubated with anti-DIG alkaline phosphatase-conjugated FAB-antibody, and washed according to the manufacturer's instructions. Luminescence was recorded on X-Ray films (Kodak) for 4 hours.
(TIF)

**S1 Table. Primers used in this study.**
(TIF)

**S1 Raw images. Raw images of the western blots.**
(PDF)

## Author Contributions

**Conceptualization:** Caterina Guzmán-Verri.

**Data curation:** Amanda Castillo-Zeledón, Nazareth Ruiz-Villalobos, Pamela Altamirano-Silva, Carlos Chacón-Díaz, Elías Barquero-Calvo.

**Formal analysis:** Amanda Castillo-Zeledón, Nazareth Ruiz-Villalobos, Pamela Altamirano-Silva, Carlos Chacón-Díaz, Elías Barquero-Calvo, Esteban Chaves-Olarte, Caterina Guzmán-Verri.

**Funding acquisition:** Elías Barquero-Calvo, Esteban Chaves-Olarte, Caterina Guzmán-Verri.

**Investigation:** Amanda Castillo-Zeledón, Nazareth Ruiz-Villalobos, Pamela Altamirano-Silva, Carlos Chacón-Díaz, Elías Barquero-Calvo, Esteban Chaves-Olarte, Caterina Guzmán-Verri.

**Methodology:** Nazareth Ruiz-Villalobos, Pamela Altamirano-Silva, Carlos Chacón-Díaz, Elías Barquero-Calvo, Esteban Chaves-Olarte.

**Project administration:** Caterina Guzmán-Verri.

**Resources:** Carlos Chacón-Díaz, Elías Barquero-Calvo, Esteban Chaves-Olarte, Caterina Guzmán-Verri.

**Supervision:** Carlos Chacón-Díaz, Elías Barquero-Calvo, Esteban Chaves-Olarte, Caterina Guzmán-Verri.

**Validation:** Amanda Castillo-Zeledón, Carlos Chacón-Díaz, Elías Barquero-Calvo, Esteban Chaves-Olarte, Caterina Guzmán-Verri.

**Visualization:** Amanda Castillo-Zeledón.

**Writing – original draft:** Amanda Castillo-Zeledón, Caterina Guzmán-Verri.

**Writing – review & editing:** Amanda Castillo-Zeledón, Nazareth Ruiz-Villalobos, Pamela Altamirano-Silva, Carlos Chacón-Díaz, Elías Barquero-Calvo, Esteban Chaves-Olarte, Caterina Guzmán-Verri.

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
