## [Decision Letter · Decision Letter 0]

5 Mar 2021

PONE-D-21-03810

A *Sinorhizobium meliloti* and *Agrobacterium tumefaciens* ExoR ortholog is not crucial for *Brucella abortus* virulence.

PLOS ONE

Dear Caterina,

Two external reviewers have evaluated your manuscript. Although both acknowledge the importance of  the information provided in terms of  improving our understanding of  the regulatory networks that control gene expression in *Brucella*, both reviewers also raise numerous points that need to be addressed/clarified before the manuscript will be considered suitable for publication in *PLOS ONE*. Thus, I am going to ask that you submit a revised version of  the manuscript that adequately and appropriately addresses all of  the concerns/issues raised by both of  these reviewers.

I look forward to receiving your revised manuscript!

Sincerely,

Marty Roop

Academic Editor

*PLOS ONE*

Journal Requirements:

2) Thank you for stating the following in the Acknowledgments Section of your manuscript:

[This work was supported by Fondos del Sistema FEES/CONARE [02-2020 to C.GV], Fondos FIDA, Universidad Nacional [SIA 0047-17 to C. G-V], Espacio Universitario de Estudios Avanzados, UCREA [B8762] from the presidency of University of Costa Rica, and the Vice Presidency for Research, University of Costa Rica [C0456 to E.C-O]]

 [The funders had no role in study design, data collection and analysis, decision to

publish, or preparation of the manuscript]

3) PLOS ONE now requires that authors provide the original uncropped and unadjusted images underlying all blot or gel results reported in a submission’s figures or Supporting Information files. This policy and the journal’s other requirements for blot/gel reporting and figure preparation are described in detail at https://journals.plos.org/plosone/s/figures#loc-blot-and-gel-reporting-requirements and https://journals.plos.org/plosone/s/figures#loc-preparing-figures-from-image-files. When you submit your revised manuscript, please ensure that your figures adhere fully to these guidelines and provide the original underlying images for all blot or gel data reported in your submission. See the following link for instructions on providing the original image data: https://journals.plos.org/plosone/s/figures#loc-original-images-for-blots-and-gels.

4) We note that you have included the phrase “data not shown” in your manuscript. Unfortunately, this does not meet our data sharing requirements. PLOS does not permit references to inaccessible data. We require that authors provide all relevant data within the paper, Supporting Information files, or in an acceptable, public repository. Please add a citation to support this phrase or upload the data that corresponds with these findings to a stable repository (such as Figshare or Dryad) and provide and URLs, DOIs, or accession numbers that may be used to access these data. Or, if the data are not a core part of the research being presented in your study, we ask that you remove the phrase that refers to these data.

Reviewers' comments:

Reviewer's Responses to Questions

**Comments to the Author**

1. Is the manuscript technically sound, and do the data support the conclusions?

Reviewer #1: Partly

Reviewer #2: Yes

2. Has the statistical analysis been performed appropriately and rigorously? 

Reviewer #1: I Don't Know

Reviewer #2: Yes

3. Have the authors made all data underlying the findings in their manuscript fully available?

Reviewer #1: Yes

Reviewer #2: Yes

4. Is the manuscript presented in an intelligible fashion and written in standard English?

Reviewer #1: Yes

Reviewer #2: Yes

5. Review Comments to the Author

Reviewer #1: Guzman-Verri and collaborators made an exoR mutant, ExoR being a strong candidate to regulate BvrS-BvrR two component system, according to data available in S. meliloti and A. tumefaciens. Surprisingly, an exoR mutant is not sensitive to polymyxin B and exoR deletion does not affect virulence in two cellular models and a mice model of infection. These data suggest that ExoR is not a functional ortholog of ExoR from S. meliloti and A. tumefaciens, which is interesting. Surprisingly, targets of the BvrS-BvrR two component system are affected by ExoR, suggesting that ExoR plays an unexpected (and unclear) role in controlling this system.

Major comments

Fig 1B : controls with bvrR and bvrS mutants must be added, to confirm that the serum sensitivity test is performed as expected

Calling the exoR mutant is maybe not appropriate, since a fraction of the protein is probably still present. Thus the authors should rename their mutant (maybe it is not a "null" mutant) and mention, in the Discussion, that they cannot exclude that the remaining protein segment kept some of the functions of the native ExoR.

Fig 2 : the data show that exoR plays a role in the control of BvrR phosphorylation level and, consequently, on the VirB8, Omp25 and VjbR anundance. A complementation should be performed to ensure that these phenotypes are dependent on the absence of ExoR.

In the Material and Methods, the cultures used for experiments like serum/PMB sensitivity, carbohydrate assimilation patterns, infections and western blot, should be more precisely described (medium, culture phase, temperature, agitation)

Minor comments

line 24 "The evidence" is rather unclear, please be more precise

line 26 "no evident effects of" by "no evident effects of exoR deletion on"

line 35 replace "preferentially" by "it preferentially"?

Fig 1CD These experiments (according to lines 291-297) have been performed in TSB, not in minimal medium (MM), is it possible that ExoR is not needed in these conditions? Serum sensitivity assay were performed on bacteria at a pH of about 7.3 since they are in PBS, again it is not a condition in which BvrR phosphorylation should occur, according to Altamirano-Silva et al (2018). I guess that in the initial model of the authors, these conditions represent those in which absence of BvrS repression by ExoR would have been found, if it is the case, this should be indicated. In the same vein, if ExoR function is to repress BvrS, then in the absence of ExoR, the BvrS-BvrR response is constitutively active, and the mutant is not different from the wild type? In this case, the absence of phenotype does indicate that ExoR "function seems different" (line 28)

line 118 In this paragraph, sensitivity to polymyxin B and serum has been tested, but it does not confirm that envelope integrity is preserved in the exoR mutant. There may be aspects of the envelope that are altered but do not lead to either serum or polymyxin B sensitivity.

line 137 "to time growth" should become "to time of growth"

line 141 BvrR instead of BvR

Fig 2 According to Altamirano-Silva et al (2018), BvrR should be phosphorylated at acidic pH in MM, thus why is it phosphorylated in TSB at (I guess) neutral pH? In Fig 2 as well, the lanes should be annotated clearly (exoR or WT under each lane). Late log and mid-log should also be indicated in this figure. Is Omp19 really a loading control? (in other words, were all labelings performed on the same membrane? probably not, thus Omp19 is not a loading control)

Line 170 Add a "." after infection.

Line 172 Replace UFC by CFU. Same remark for Fig 3C. Also indicate that you count CFU per gram of spleen (I guess)

line 189 Replace "is" by "are"

lines 257-259 If the primers exoR-F1 and exoR-R4 are used to confirm exoR deletion, the authors distinguish these deletion events from the ectopic integration of the plasmid in the B. abortus genome. Thus Southern blot is crucial and its description (lines 272-289) should include the nature of the probe used. What fragment is amplified by the bruabI0884.3 and bruabI0884.5 primers?

line 266 replace "Biomeriux" by "Biomerieux"

Reviewer #2: The manuscript by Castillo-Zeledon and colleagues describes the identification and characterization of an ExoR ortholog in Brucella abortus. Previous literature in other members of the Alphaproteobacteria have defined the role of EoxR in the regulatory pathways of important two-component systems, and in B. abortus, the orthologous system is called BvrR/BvrS. To date, no ExoR protein has been identified and characterized in Brucella species, but the authors have identified a putative ExoR protein. Using an exoR deletion strain, the authors demonstrate that deletion of exoR does not affect normal growth of the bacteria in vitro, and the exoR deletion strain also does not exhibit any reduced survival in the presence of membrane perturbation. The authors show that deletion of exoR leads to aberrant production of important proteins, such as VirB8 and Omp25, and VjbR; however, there is no defect in the ability of the exoR deletion strain to infect and survive in cells or animals (i.e., mice).

Overall, the conclusions are well supported by the data, and the experiments were performed soundly. While he findings do not point to the importance of ExoR in virulence in a mouse model, the data still provide important insight into a previously unknown protein in Brucella. Nonetheless, there are a few minor issues the authors need to address:

-Line 35: "where preferentially" should probably be "where it preferentially"

-Line 123: bvrR and bvrS need to be italicized.

-In several places the term "Western Blot" is used, but there is no need to capitalize this term. Please see PMID: 27893304. The specific places in the manuscript using "Western Blot" are lines 129, 139-140, 146, and 150.

-The labeling of Figure 2 makes interpretation of the data very difficult. It is stated that odd lanes are the exoR mutant and even lanes are the wild-type, but the lanes are not labeled with numbers. In fact, numbes may further add to confusion, and it would help if the authors used "WT" and "exoR" to depict which lanes represent which strains.

-Line 194: "identity level" sounds a bit odd, and it might be better to state "high degree of identity" instead.

-Lines 205-206 present a paradox. It is stated that "expression of these proteins is independent of ExoR," and then it is stated that "the results suggest that ExoR is related to the BvrR phosphorylation state." Perhaps clarifying which "proteins" line 205 is referring to would help alleviate the confusion, but as is, it is not clear if the authors are stating the conclusion that EoxR is or is not linked to the BvrR/BvrS systems. Some additional clarification on this point could be helpful.

6. PLOS authors have the option to publish the peer review history of their article (what does this mean?). If published, this will include your full peer review and any attached files.

Reviewer #1: No

Reviewer #2: No

---

## [Author Response · Author response to Decision Letter 0]

8 Jun 2021

THIS INFO IS INCLUDED IN THE COVER LETTER IN A TABLE FORMAT. WE THANK THE REVIEWERS FOR THEIR VALUABLE COMMENTS.

 The manuscript, as well as the file naming, has been reviewed accordingly

 [The funders had no role in study design, data collection, and analysis, decision to

publish, or preparation of the manuscript]

The funding information was removed from the Acknowledgments section.

The Funding Statement should read as follows:

“This work was supported by Fondos del Sistema FEES/CONARE [02-2020, 0652-19 to C. G-V], Fondos FIDA, Universidad Nacional [SIA 0047-17 to C. G-V], Espacio Universitario de Estudios Avanzados, UCREA [B8762] from the presidency of University of Costa Rica, and the Vice Presidency for Research, University of Costa Rica [C0456 to E.C-O].

The funders had no role in study design, data collection, and analysis, decision to

publish, or preparation of the manuscript.”

3) PLOS ONE now requires that authors provide the original uncropped and unadjusted images underlying all blot or gel results reported in a submission’s figures or Supporting Information files. This policy and the journal’s other requirements for blot/gel reporting and figure preparation are described in detail at https://journals.plos.org/plosone/s/figures#loc-blot-and-gel-reporting-requirements and https://journals.plos.org/plosone/s/figures#loc-preparing-figures-from-image-files. When you submit your revised manuscript, please ensure that your figures adhere fully to these guidelines and provide the original underlying images for all blot or gel data reported in your submission. See the following link for instructions on providing the original image data: https://journals.plos.org/plosone/s/figures#loc-original-images-for-blots-and-gels.

This info is in file S1_raw_image

4) We note that you have included the phrase “data not shown” in your manuscript. Unfortunately, this does not meet our data sharing requirements. PLOS does not permit references to inaccessible data. We require that authors provide all relevant data within the paper, Supporting Information files, or in an acceptable, public repository. Please add a citation to support this phrase or upload the data that corresponds with these findings to a stable repository (such as Figshare or Dryad) and provide and URLs, DOIs, or accession numbers that may be used to access these data. Or, if the data are not a core part of the research being presented in your study, we ask that you remove the phrase that refers to these data. The phrase “data not shown” was removed (page 5, line 107). The Southern blot result was added as a supplementary figure S1_Fig: Mutant construction and Southern blot. 

Reviewer #1 

Major comments

Fig 1B: controls with bvrR and bvrS mutants must be added, to confirm that the serum sensitivity test is performed as expected.

ANSWER: We followed the reviewer's suggestion and repeated the assays including the corresponding mutant controls. We also incubated the strains for a longer period with serum (90 min). This is now shown in Figure 1B. The figure legend was modified accordingly and includes a suggestion from the other reviewer, indicating that sensitivity to polymyxin B and serum does not confirm cell envelope integrity. 

The figure legend is as follows: (page 7, line 141-151) 

“Fig 1. The exoR mutation does not alter bacterial growth, non-immune human serum treatment or resistance to polymyxin B. (A) B. abortus 2308W (WT) and the exoR mutant (exoR) were grown in TSB at 37⁰C and the absorbance was measured at 420 nm at the indicated times. (B) The indicated strains B. abortus 2308W (WT), exoR, bvrS, and bvrR mutants, were exposed to non-immune human serum at 37⁰C for 90 min. These experiments are representative of at least three performed. (C) and (D) The indicated strains, B. abortus 2308W (WT), exoR, bvrS, and bvrR mutants, were exposed to increasing concentrations of polymyxin B for 48 hours at 37⁰C, at pH 7 and pH 6, respectively. The minimum inhibitory concentrations shown are representative of at least three experiments performed. The bvrS and bvrR mutant strains were used as susceptibility controls. *, P < 0,05 (Anova and Kruskal-Wallis test for multiple comparisons).

Calling the exoR mutant is maybe not appropriate, since a fraction of the protein is probably still present. Thus the authors should rename their mutant (maybe it is not a "null" mutant) and mention, in the Discussion, that they cannot exclude that the remaining protein segment kept some of the functions of the native ExoR. 

ANSWER: We agree with the reviewer and renamed the strain “B. abortus exoR mutant” instead of null mutant in the entire document. 

Also, we included in the discussion the next paragraph: 

(page 14, line 316-320) 

“Since the introduced deletion in exoR did not remove the gene entirely, we cannot exclude that the remaining protein segment kept some of the functions of the native ExoR. More studies are needed to understand the role of the conserved ExoR, not only in B. abortus but also in other members of the genus, as well as other members of the Alphaproteobacteria.”

Fig 2: the data show that exoR plays a role in the control of BvrR phosphorylation level and, consequently, on the VirB8, Omp25 and VjbR anundance. A complementation should be performed to ensure that these phenotypes are dependent on the absence of ExoR. 

ANSWER: We understand the reviewer's opinion. The results show punctual changes in BvrR phosphorylation and protein expression at specific points during bacterial growth in TSB pH 7. In this revised version, new experiments show that under conditions that promote BvrR/BvrS activation (MM and pH5.0) the trend is similar, i.e no drastic changes in protein expression as compared to the wild type, but rather subtle ones related to the protein expression dynamics. Although the idea of a complemented strain expressing exoR sounds the reasonable next step, it is our experience that complementation experiments related to gene regulation networks and involving expression of proteins related to the bacterial membrane usually do not give clear answers, but rather inconclusive ones. The lack of proper molecular biology tools proven to work in Brucella, where gene and protein expression can be accurately controlled, also precluded our intention to perform such experiments. Our opinion is that the trials that indicate a clear phenotype and relationship with bacteria virulence are the in vivo and ex vivo models using mice and cell infections, where the mutant behaved just like the wild-type strain. Therefore, we conclude that under these conditions, a complemented strain would not add any significant value to our findings.

In the Material and Methods, the cultures used for experiments like serum/PMB sensitivity, carbohydrate assimilation patterns, infections, and western blot, should be more precisely described (medium, culture phase, temperature, agitation) 

ANSWER:We followed the reviewer´s suggestion and incorporated a more precise description for the indicated experiments in the methods section: 

The phrase:

“B. abortus 2308W (WT Brucella strain, virulent, smooth LPS, NaIr) [50] or B. abortus 2308W ΔexoR mutant strain (this study) were maintained and grown in standard Tryptic Soy Broth (TSB) at 37⁰C.” 

Was replaceD for: (page 15; line 327-330) 

“B. abortus 2308W (WT Brucella strain, virulent, smooth LPS, NaIr) [52] or B. abortus 2308W exoR mutant strain (this study) were maintained and grown at 37⁰C with agitation (200 rpm) in standard Tryptic Soy Broth (TSB) at pH 7.0 or minimal medium (33 mM KH2PO4, 60.3 mM K2HPO4, and 0.1% yeast extract) at pH 5.0 (adjusted with citric acid).”

The phrase 

“Essentially, exponentially growing bacteria of each strain were adjusted to 5x106 CFU/mL in TSB. 

Was replaced for: (Page 17, line 375-377)

“Essentially, both strains, mutant, and wild-type, were grown until exponential phase in TSB at pH 7, with agitation of 200 rpm. Bacteria of each strain were adjusted to 5x106 CFU/mL in TSB. 

The phrase: 

“Exponentially growing bacteria were adjusted to 104 CFU/ml in PBS and dispensed in duplicate in microtiter plates (200 µl per well) containing fresh human serum (400 µl/well). After 45 min of incubation at 37°C, 100 µl was plated on tryptic soy agar.” 

Was replaced for: (page 17, line 383-385) 

 “Exponentially growing bacteria grown in TSB at pH 7, with the agitation of 200 rpm, were adjusted to 104 CFU/ml in PBS and dispensed in duplicate in microtiter plates (200 µl per well) containing fresh human serum (400 µl/well). After 90 min of incubation at 37°C, 100 µl was plated on tryptic soy agar.”

The phrase:

 “Bacteria were concentrated by centrifugation, resuspended in Laemmli sample buffer, and heated at 100°C for 20 min.” 

Was replaced for: (page 18, line 394-396) 

“Bacteria grown to exponential phase in TSB at pH 7, with the agitation of 200 rpm, were concentrated by centrifugation, resuspended in Laemmli sample buffer, and heated at 100°C for 20 min.

The phrase 

“BALB/c female (18-24 grams) mice were intraperitoneally (i.p.) inoculated with 106 UFC of either B. abortus 2308W or B. abortus 2308W ΔexoR mutant and sacrificed at 3 and 9 weeks post-infection.” 

Was replaced for: (page 20, line 436-439) 

“BALB/c female (18-24 grams) mice were intraperitoneally (i.p.) inoculated with 106 UFC of either B. abortus 2308W or B. abortus 2308W exoR mutant at exponential phase grown in TSB at pH 7, with the agitation of 200 rpm. Mice were sacrificed at 3- and 9-weeks post-infection.”

Minor comments 

line 24 "The evidence" is rather unclear, please be more precise 

ANSWER:The phrase:

“The evidence reveals that ExoR is related to the BvrR phosphorylation state, and the absence of ExoR has different impacts on known BvrR/BrvS gene targets. 

was modified according to the reviewer's suggestion as follows (page 2, line 24-27):

Our findings indicate that ExoR is related to the BvrR phosphorylation state, and to the expression of known BvrR/BrvS gene targets, such as virB8, vjbR, and omp25 when grown in rich medium or starving conditions. 

line 26 "no evident effects of" by "no evident effects of exoR deletion on" 

ANSWER:The phrase;

“Despite this, we found no evident effects of in vitro growth, intracellular replication, or infectivity in a mice model”

was modified according to the reviewer's suggestions, including the renaming of the exoR mutant, as follows: (page 2, line 27-29).

“Despite this, the exoR mutant strain showed no significant differences as compared to the wild type strain, related to resistance to polymyxin B or human non-immune serum, intracellular replication, or infectivity in a mice model.”

line 35 replace "preferentially" by "it preferentially"? 

ANSWER:The suggestion was included as follows: 

(page 2, line 37) Brucella abortus infects bovines, where it preferentially replicates in reproductive organs, causing abortion, infertility, decreased milk production, reproductive failure, and epididymitis [2,3]

Fig 1CD These experiments (according to lines 291-297) have been performed in TSB, not in minimal medium (MM), is it possible that ExoR is not needed in these conditions? Serum sensitivity assay were performed on bacteria at a pH of about 7.3 since they are in PBS, again it is not a condition in which BvrR phosphorylation should occur, according to Altamirano-Silva et al (2018). I guess that in the initial model of the authors, these conditions represent those in which absence of BvrS repression by ExoR would have been found, if it is the case, this should be indicated. In the same vein, if ExoR function is to repress BvrS, then in the absence of ExoR, the BvrS-BvrR response is constitutively active, and the mutant is not different from the wild type? In this case, the absence of phenotype does indicate that ExoR "function seems different" (line 28)

ANSWER: We thank the reviewer for this comment. To clarify, BvrR phosphorylation does occur under TSB and neutral pH conditions. This was shown by Altamirano-Silva et al (2018), Figure 2 where lysates from bacteria grown at pH 7 in TSB were used as BvrR-phosphorylation control, as well as in Figure 3. On the other hand, when bacteria are transferred to an acidic pH and MM, Altamirano-Silva et al (2018) observed a transient increase in BvrR phosphorylation as compared to bacteria transferred into TSB at neutral pH. This effect is shown in Altamirano-Silva et al (2018) Figure 6.

Following the reviewer´s suggestion and to assess if ExoR is not needed after exposure to minimal medium and pH 5, we performed the serum sensitivity and the polymyxin resistance assays using wild-type and exoR mutant strains grown to exponential phase and incubated in minimal medium (MM) at pH 5.0 or rich medium (TSB) at pH 7.0 for 4 hours. We also assessed the expression of VjbR and VirB8 after exposure of the exoR mutant exponentially growing cultures, to MM pH 5 for 0.5, 2, 4, and 6 hours. The results are included as a new figure, Fig 4 and modifications to the Methods, results, and discussion section were introduced accordingly (results section (page 10-11, line 206-237), Fig 4, as well as in the discussion section (page 14, line 302-308). 

Regarding the second question, we are not sure if we understand the reviewer´s reasoning. As shown in fig 1, the BvrR phosphorylation pattern is a dynamic process, occurring both in the wild-type and the exoR mutant. The main difference in this pattern between both strains is that the BvrR-P signal faded faster in the exoR mutant, as compared to the wild-type strain. This is interpreted as that indeed, ExoR is related to BvrR phosphorylation dynamics and hence, to some BvrR regulation targets. The reason for this observation remains elusive and is not relevant in terms of B. abortus virulence, at least in a mice model. The phrase “function seems different” pointed out by the reviewer, is related to ExoR orthologs. To clarify those points, we made the following modifications:

Starting at line 24 of the previous version:

“The evidence reveals that ExoR is related to the BvrR phosphorylation state, and the absence of ExoR has different impacts on known BvrR/BrvS gene targets. Despite this, we found no evident effects of in vitro growth, intracellular replication, or infectivity in a mice model. ExoR in B. abortus is related to BvrR/BvrS as observed in other Rhizobiales; however, its function seems different from that observed for its orthologs described in A. tumefaciens and S. meliloti.”

It is now replaced by the following: (page 2, line: 24-31)

“Our findings indicate that ExoR is related to the BvrR phosphorylation state, and to the expression of known BvrR/BrvS gene targets, such as virB8, vjbR, and omp25 when grown in rich medium or starving conditions. Despite this, the exoR mutant strain showed no significant differences as compared to the wild-type strain, related to resistance to polymyxin B or human non-immune serum, intracellular replication, or infectivity in a mice model. ExoR in B. abortus is related to BvrR/BvrS as observed in other Rhizobiales; however, its function seems different from that observed for its orthologs described in A. tumefaciens and S. meliloti.”

Starting at line 206, Discussion section in the previous version:

“The results suggest that ExoR is related to the BvrR phosphorylation state. The data presented here show that exoR plays a role in the control of BvrR phosphorylation level. The exoR mutant showed a gradual fade of BvrR-P signal during bacterial growth, which occurs faster than in the wild-type strain. There is no direct evidence linking the phosphorylation of the response regulator and ExoR in other bacterial RSI switches; but, in A. tumefaciens and S. meliloti ExoR absence induces a phenotype similar to that of bacteria with a constitutive TCS. This does not seem to be the case in B. abortus 2308.”

It is now modified as follows (page 13, line 286-294, Discussion section): 

“This suggests that their expression is independent of ExoR, but possible posttranslational modifications (including phosphorylation and dephosphorylation) might not, since the BvrR phosphorylation dynamics was different as compared to the wild-type strain. The data presented here show that exoR plays a role in the control of BvrR phosphorylation level. The exoR mutant showed a gradual fade of BvrR-P signal during bacterial growth, which occurs faster than in the wild-type strain. There is no direct evidence linking the phosphorylation of the response regulator and ExoR in other bacterial RSI switches; but, in A. tumefaciens and S. meliloti ExoR absence induces a phenotype similar to that of bacteria with a constitutive TCS. This does not seem to be the case in B. abortus 2308W.”

line 118 In this paragraph, sensitivity to polymyxin B and serum has been tested, but it does not confirm that envelope integrity is preserved in the exoR mutant. There may be aspects of the envelope that are altered but do not lead to either serum or polymyxin B sensitivity. 

ANSWER:We agree with the reviewer´s comment. The subtitle “ExoR is not related to cell envelope integrity.”

 was modified to (page 7, line 153-154)

“The exoR mutation does not affect B. abortus resistance to polymyxin B and non-immune human serum.” 

Also, the wording of the following paragraph was amended (page 7, line 155-161). 

“The B. abortus LPS smooth phenotype was not altered by the exoR mutation, as judged by the acriflavine agglutination test. Exposure to non-immune human serum and increasing polymyxin B concentrations at pH 7 and pH 6 did not reveal differences between the wild-type and the exoR mutant strain (Figs 1B, C and D). As expected, the bvrR and bvrS mutants, included as controls, were sensitive to polymyxin B and non-immune human serum (Fig 1C and D). Both strains showed significant differences when compared to the wild-type and the exoR mutant.“

As well as Figure 1 title: “Fig 1. The absence of ExoR does not alter bacterial growth nor membrane integrity.” 

was modified accordingly: (page 7, line 141-142)

“Fig 1. The exoR mutation does not alter bacterial growth, resistance to non-immune human serum or polymyxin B treatment.”

line 137 "to time growth" should become "to time of growth" 

ANSWER:The phrase: “Omp25 showed slight differences in expression according to time growth as reported.” 

was modified according to the reviewer's suggestion as follows: (page 9, line 190-191)

“Omp25 showed slight differences in expression according to the wild-type strain time of growth as reported [39].”

line 141 BvrR instead of BvR 

ANSWER:The phrase:

“As described above, the BvR expression was consistent between strains” is no longer in the manuscript due to the following modification: (page 8, line 166-167).

“We observed similar BvrR and BvrS expression patterns compared to the wild-type strain in the mutant strain (Fig 2).”

Fig 2 According to Altamirano-Silva et al (2018), BvrR should be phosphorylated at acidic pH in MM, thus why is it phosphorylated in TSB at (I guess) neutral pH? In Fig 2 as well, the lanes should be annotated clearly (exoR or WT under each lane). Late log and mid-log should also be indicated in this figure. Is Omp19 really a loading control? (in other words, were all labelings performed on the same membrane? probably not, thus Omp19 is not a loading control) 

ANSWER:As mentioned above, BvrR does phosphorylate in TSB at neutral pH (Altamirano-Silva et al., 2018). 

We thank the reviewer´s observation in Fig 2. Lanes are now annotated clearly and each growth phase assessed is indicated. The western blot was performed in different gels, the loading control for each gel was added to the figure. In order not to overload the figure, we divided it into two: one that evaluated the two-component system and another one that evaluates three proteins under BvrRBvrS regulation (Fig 2 and 3). Due to this new structure, the results section where the western blot is explained was modified accordingly. (page 8-9, line 163-204)

The subtitle “ExoR affects the expression of proteins related to BvrR/BvrS.” 

Is now replaced for: (page 8, line 163-164)

“ExoR impacts the expression dynamics of proteins related to BvrR/BvrS in nutrient-rich conditions.”

Regarding to the loading control, labeling of the loading control was performed in the same membrane for each of the protein tested. This is clarified in the materials and methods section (page 19, line 407-409) and in Fig 2 and 3.

Line 170 Add a "." after infection.

ANSWER: The "." was added (page 12, line 255).

Line 172 Replace UFC by CFU. Same remark for Fig 3C. Also indicate that you count CFU per gram of spleen (I guess) 

ANSWER:The phrase: 

“Mice were intraperitoneally inoculated with 106 UFC of either B. abortus 2308W or B. abortus 2308W ΔexoR mutant. Spleen counts were determined at 3 and 9 weeks post-infection.” 

Was replaced for: (page12, line 252-255, and figure 5C).

“Mice were intraperitoneally inoculated with 106 CFU of either B. abortus 2308W or B. abortus 2308W exoR mutant. CFU per gram of spleen were determined at 3 and 9 weeks post-infection.” 

line 189 Replace "is" by "are" 

ANSWER:The phrase: 

“Proteins whose gene expression is under the control of BvrR/BvrS showed differentiated expression in the exoR mutant as compared to the wild-type strain.”

Was modified for: (page 12, line 269-271)

“Proteins whose gene expression are under the control of BvrR/BvrS [22,28] showed differentiated expression in the exoR mutant as compared to the wild-type strain.” 

lines 257-259 If the primers exoR-F1 and exoR-R4 are used to confirm exoR deletion, the authors distinguish these deletion events from the ectopic integration of the plasmid in the B. abortus genome. Thus Southern blot is crucial and its description (lines 272-289) should include the nature of the probe used. What fragment is amplified by the bruabI0884.3 and bruabI0884.5 primers?

ANSWER:For better understanding, we included a schematic representation of the in-frame deletion strategy used to construct the exoR mutant and the Southern blot result as a supplementary figure (S1_Fig.) (Page 5, line 111-139). And wording was improved in Material and Methods (page 16-17, line 359-371).

line 266 replace "Biomeriux" by "Biomerieux" 

ANSWER:The word Biomerieux was corrected (page 16, line 351).

Reviewer #2 

-Line 35: "where preferentially" should probably be "where it preferentially" 

ANSWER:The suggestion was included as follows: 

(page 2, line 37) Brucella abortus infects bovines, where it preferentially replicates in reproductive organs, causing abortion, infertility, decreased milk production, reproductive failure, and epididymitis.

-Line 123: bvrR and bvrS need to be italicized. 

ANSWER:Both words were italicized (page 7, line 158).

-In several places, the term "Western Blot" is used, but there is no need to capitalize this term. Please see PMID: 27893304. 

ANSWER:The specific places in the manuscript using "Western Blot" are lines 129, 139-140, 146, and 150. The term “Western Blot” was replaced by “western blot” in the manuscript.

-The labeling of Figure 2 makes interpretation of the data very difficult. It is stated that odd lanes are the exoR mutant and even lanes are the wild-type, but the lanes are not labeled with numbers. In fact, numbes may further add to confusion, and it would help if the authors used "WT" and "exoR" to depict which lanes represent which strains. 

ANSWER:Indeed, we agree with the reviewer. For the labeling of the figure, we followed the reviewer's suggestion. Lanes were annotated clearly, the growth phase was indicated and as the western blots were performed in different gels, the loading control for each gel was added to the figure. In order not to overload the figure, we divided it into two figures: one that evaluated the two-component system and another one that evaluates three proteins under BvrR/BvrS regulation (Fig 2 and 3). 

The results section where the western blots are explained was modified accordingly (page 8-9, line 165-206).

The subtitle “ExoR affects the expression of proteins related to BvrR/BvrS.”, 

was replaced for: (page 8, line 163-164)

ExoR impacts the expression dynamics of proteins related to BvrR/BvrS in nutrient rich conditions.

-Line 194: "identity level" sounds a bit odd, and it might be better to state "high degree of identity" instead. 

ANSWER:The suggestion was included as follows: 

(page 13, line 277-279)

Considering the high degree of identity between the BvrR, BvrS, and ExoR homologs, these results were unexpected. 

-Lines 205-206 present a paradox. It is stated that "expression of these proteins is independent of ExoR," and then it is stated that "the results suggest that ExoR is related to the BvrR phosphorylation state." Perhaps clarifying which "proteins" line 205 is referring to would help alleviate the confusion, but as is, it is not clear if the authors are stating the conclusion that EoxR is or is not linked to the BvrR/BvrS systems. Some additional clarification on this point could be helpful. 

ANSWER:We agreed with the reviewer´s comment, so the wording was changed as follows:

The phase: 

“Therefore, the expression of these proteins is independent of ExoR. The results suggest that ExoR is related to the BvrR phosphorylation state.”

Was replaced for: (page 13, lines 287-293).

“Moreover, BvrR and BvrS did not show differences in their expression compared to the wild-type. This suggests that their expression is independent of ExoR, but possible posttranslational modifications (including phosphorylation and dephosphorylation) might not, since the BvrR phosphorylation dynamics was different as compared to the wild-type strain. The exoR mutant showed a gradual fade of BvrR-P signal during bacterial growth, which occurs faster than in the wild-type strain.”

---

## [Decision Letter · Decision Letter 1]

30 Jun 2021

A *Sinorhizobium melilot*i and *Agrobacterium tumefaciens* ExoR ortholog is not crucial for *Brucella abortus* virulence.

PONE-D-21-03810R1

Dear Caterina,

We’re pleased to inform you that your manuscript has been judged scientifically suitable for publication and will be formally accepted for publication once it meets all outstanding technical requirements.

Sincerely,

Marty Roop

Academic Editor

*PLOS ONE*

Additional Editor Comments (optional):

Reviewers' comments:

Reviewer's Responses to Questions

**Comments to the Author**

1. If the authors have adequately addressed your comments raised in a previous round of review and you feel that this manuscript is now acceptable for publication, you may indicate that here to bypass the “Comments to the Author” section, enter your conflict of interest statement in the “Confidential to Editor” section, and submit your "Accept" recommendation.

Reviewer #1: All comments have been addressed

Reviewer #2: All comments have been addressed

2. Is the manuscript technically sound, and do the data support the conclusions?

Reviewer #1: Yes

Reviewer #2: Yes

3. Has the statistical analysis been performed appropriately and rigorously? 

Reviewer #1: Yes

Reviewer #2: Yes

4. Have the authors made all data underlying the findings in their manuscript fully available?

Reviewer #1: Yes

Reviewer #2: Yes

5. Is the manuscript presented in an intelligible fashion and written in standard English?

Reviewer #1: Yes

Reviewer #2: Yes

6. Review Comments to the Author

Reviewer #1: The authors correctly addressed all my comments, including those related to a precise description of their experiments.

Reviewer #2: The authors have carefully addressed the previous comments, and there are no additional concerns.

7. PLOS authors have the option to publish the peer review history of their article (what does this mean?). If published, this will include your full peer review and any attached files.

Reviewer #1: No

Reviewer #2: No

---

## [Editor Report · Acceptance letter]

5 Aug 2021

PONE-D-21-03810R1 

A *Sinorhizobium meliloti* and *Agrobacterium tumefaciens* ExoR ortholog is not crucial for *Brucella abortus* virulence. 

Dear Dr. Guzman-Verri:

I'm pleased to inform you that your manuscript has been deemed suitable for publication in PLOS ONE. Congratulations! Your manuscript is now with our production department. 

Kind regards, 

on behalf of

Dr. Roy Martin Roop II 

Academic Editor

PLOS ONE